# Macrophage-Derived Factors with the Potential to Contribute to the Pathogenicity of HIV-1 and HIV-2: Roles of M-CSF and CXCL7

**DOI:** 10.3390/ijms26115028

**Published:** 2025-05-23

**Authors:** Chunling Gao, Joseph Kutza, Weiming Ouyang, Tobias A. Grimm, Karen Fields, Carla S. R. Lankford, Franziska Schwartzkopff, Mark Paciga, Ana Machuca, Linda Tiffany, Tzanko Stantchev, Kathleen A. Clouse

**Affiliations:** 1Division of Pharmaceutical Quality Research Four (DPQR-IV), Office of Pharmaceutical Quality Research, Center for Drug Evaluation and Research, U.S. Food and Drug Administration, 10903 New Hampshire Avenue, Silver Spring, MD 20993, USAkcstrebel@verizon.net (K.A.C.); 2Division of Pharmaceutical Quality Research Three (DPQR-III), Office of Pharmaceutical Quality Research, Center for Drug Evaluation and Research, U.S. Food and Drug Administration, 10903 New Hampshire Avenue, Silver Spring, MD 20993, USA; 3Laboratory of Molecular Virology, Division of Emerging and Transfusion Transmitted Diseases (DETTD), Center for Biologics Evaluation and Research (CBER), U.S. Food and Drug Administration, 10903 New Hampshire Avenue, Silver Spring, MD 20993, USA

**Keywords:** HIV-1, HIV-2, M-CSF, CXCL7, macrophage, replication, chemokine

## Abstract

Human immunodeficiency virus (HIV) type 2 (HIV-2) is less pathogenic than HIV-1. However, the factors responsible for the differences in pathogenicity are still not well defined. To investigate this issue, we performed infection of primary human monocyte-derived macrophages (MDMs) with individual HIV-1 or HIV-2 strains and compared the levels of M-CSF, a cytokine shown to promote HIV-1 infection and replication in our previous studies, and CXCL7, a chemokine identified as being expressed at levels correlated with HIV type by our preliminary gene-expression analysis. We tested several HIV-2 isolates able to replicate in human MDMs and observed that all of them induced the production of M-CSF at high levels similar to those previously established for HIV-1 infection. In addition, the production of M-CSF in MDMs infected with HIV-1 or HIV-2 isolates correlated with the extent of virus replication. In contrast to M-CSF, the chemokine CXCL7 was differentially expressed between MDMs infected with HIV-1 or HIV-2 isolates, as revealed by qPCR and ELISA testing. Together, these results suggest that M-CSF induction may play similar roles in promoting the replication of HIV-1 and HIV-2, while differential regulation of chemokine expression may be an important factor contributing to the differential pathogenicity of the two HIV subtypes.

## 1. Introduction

Acquired immunodeficiency syndrome (AIDS) can develop following infection with HIV-1 [1,2] or HIV-2 [3,4,5] or following co-infection with HIV-1 and HIV-2 [6,7,8]. Both HIV-1 and HIV-2 arose following several zoonotic cross-species transmissions of simian immunodeficiency viruses (SIV), which include the chimpanzee virus SIV_cpz_, the predecessor of HIV-1 groups M and N; the gorilla virus SIV_gor_ for HIV-1 groups O and P; and the sooty mangabey virus SIV_smm_ for HIV-2 groups A through I [3,9,10]. Although their genomic organization and clinical features are similar [9,11], HIV-2 infections are predominantly restricted to West Africa, occur less frequently, and are generally less pathogenic than HIV-1 infections [4,7,8,11]. The progression to AIDS in HIV-2-infected individuals is also reduced and/or delayed, with a corresponding mortality rate about two-thirds lower than for HIV-1. Intriguingly, there are observations suggesting that HIV-2 and HIV-1 co-infection may prolong the time to progression to AIDS compared to HIV-1 mono-infection [12,13], although this correlation was not confirmed in all related studies [14]. HIV-2 also has lower transmission rates, and while HIV-1 infection is associated with a steady decline in CD4^+^ T-lymphocyte counts, in HIV-2 infection, the decline is much slower and viremia levels are lower at all stages of the disease [15,16]. The immune response to HIV infection may be an important factor in determining the clinical course of disease [17]. More efficient innate and adaptive anti-HIV-2 immune responses appear to contribute to the observed differences from HIV-1 [3,6,8,11,17,18,19]. Cytokines are an integral part of the immune response, and a more in-depth understanding of their effects and patterns in HIV-1 and HIV-2 infections may provide an opportunity to better understand the mechanisms that play a role in virus control [17,20].

HIV initially infects CD4^+^ memory T cells and uses CCR5 as the co-receptor for entry [17,21,22]. An exponential burst of virus replication follows, and, around the time of peak viremia, significant depletion of CD4^+^ mucosal T cells occurs. In addition to CD4^+^ T lymphocytes, cells of the monocyte/macrophage lineage contribute to HIV pathogenesis at multiple points during the course of disease. Macrophages are among the first cells infected with HIV-1 [1,22,23,24]. There is also evidence that resident macrophages in multiple anatomical sites, such as the central nervous system (CNS), lymph nodes, and gut lymphoid tissue, are targets for HIV during acute infection and that the HIV proviral DNA may be integrated into the host-cell genome [25,26,27,28,29]. Macrophages are long-lived cells that are relatively resistant to the cytopathic effects of HIV-1 and can therefore function as a reservoir for HIV and thus as a vector for dissemination of the virus to other susceptible cells, including CD4^+^ T-lymphocytes [30,31,32,33,34,35]. Furthermore, HIV-infected macrophages may persist during highly active antiretroviral therapy, thus hampering virus eradication, which may thus require the development of more macrophage-specific anti-HIV therapies [25,26,28,29,36,37]. Additionally, macrophages are regulatory cells that control the pace and intensity of disease progression via the production of cytokines during the course of innate and adaptive immune responses [24,38]. Infection of macrophages by HIV-1 is clearly an important component of AIDS pathogenesis and contributes to comorbidities during HIV infection [28,34,35,38,39]. Similarly to HIV-1, HIV-2 can infect macrophages as well as T lymphocytes and utilize chemokine co-receptors for viral entry [40,41]. HIV-2 strains are generally more promiscuous than HIV-1 strains in their use of co-receptors, but CCR5 and CXCR4 appear to still be the major co-receptors for infection, and it is not clear whether use of the other co-receptors is relevant in vivo [16,42,43,44].

Systemic cytokine/chemokine release has been observed following HIV-1 infection [17,20,45]. Characterization of the cytokines/chemokines that can be induced by HIV-1 and HIV-2 is therefore important to understanding viral pathogenesis. We have previously reported that M-CSF is endogenously produced by human macrophages infected with HIV-1, as is reflected by increases at both the mRNA and protein levels [46]. Exogenous M-CSF is capable of rendering macrophages more susceptible to HIV-1 infection, most likely through its ability to enhance expression of the HIV-1 receptor and co-receptor (CD4 and CCR5, respectively) [47,48,49]. Although several reports document the ability of HIV-2 to infect human macrophages [40,50,51], in-depth studies on the impact of HIV-2 on the production of cytokines by infected macrophages have not been performed. Thus, compared to HIV-1, the role of macrophages in the HIV-2 disease process is not well delineated.

Chemokines are a group of small, soluble proteins with chemoattractant and immunomodulatory activities that belong to the cytokine family. Recruitment of macrophages and T cells to the infection site(s) by chemokines, induced following HIV infection, facilitates expansion of an ongoing infection and the establishment of virus reservoirs [52,53,54]. Recently, we reported that expression of the β-chemokine CCL2 is increased in HIV-1-infected human MDMs but decreased in HIV-2-infected MDMs when compared to uninfected controls [41]. The decrease in CCL2 expression following HIV-2 infection occurs at both the mRNA and protein levels. However, it remains to be investigated whether additional chemokine(s) can also be differentially regulated following HIV-1 or HIV-2 infection.

Our current study assessed the ability of primary or minimally passaged HIV-2 isolates to infect human MDMs in vitro and compared the expression levels of M-CSF and CXCL7 in macrophages following infection with individual HIV-1 or HIV-2 isolates.

## 2. Results

### 2.1. HIV-2 Isolates Use Various Co-Receptors for Infection

Our previous studies showed that M-CSF production can be induced by HIV-1 infection of human MDMs [46,55]. We next examined whether HIV-2 infection of MDMs could also induce M-CSF production. To this end, we used several minimally passaged HIV-2 isolates reported to use different co-receptors for their infection (Table 1) [56]. Using a GHOST cell assay [57,58], we confirmed that the HIV-2 B3-B4 and HIV-2 B7-B9 isolates utilize CCR5 as their co-receptor. The M-tropic HIV-2 Rod isolate uses CXCR4 as its co-receptor, while the HIV-2 B2 and HIV-2 B5 isolates could potentially employ CCR5, CXCR4, CCR1, and CCR2b as co-receptors (Table 1 and Figure 1).

### 2.2. MDMs Infected with HIV-1 and HIV-2 Isolates Produce Similar Levels of M-CSF

Initially, monocytes from one healthy donor were differentiated into macrophages, infected with the pleiotropic HIV-2 isolate B5, the CXCR4-tropic isolate Rod, and the CCR5-tropic HIV-2 isolates B7 and B8 and cultured for 39 days. The lab-adapted HIV-1 strain Ada served as a control. Measurement of the HIV RT activities in the culture supernatants collected from day 3 to day 39 at 6-day intervals indicated that all HIV isolates replicated in human MDMs, although their replication capabilities were different (Figure 2A). HIV-1 Ada exhibited the most robust replication, while HIV-2 isolates B8 and Rod showed low levels of replication in the human MDMs (Figure 2A). Interestingly, MDMs infected with the various HIV isolates all produced M-CSF at similar levels (Figure 2B). To further corroborate these observations, we repeated the time-course study by performing infection of MDMs derived from four additional healthy donors using the same HIV isolates. Results from the repeated experiments confirmed that the HIV isolates replicated in human MDMs to different degrees, while their replication induced the production of similar amounts of M-CSF (Figure 2C,D).

Further statistical analyses of M-CSF production at the time of peak virus replication in MDMs from 6–10 donors using one-way ANOVA with Dunnett’s test for multiple testing corrections indicated that the average levels of M-CSF expression were significantly different between the uninfected and the five HIV-infected groups (*p* = 0.0033) (Figure 3). Dunnett’s test for multiple comparisons, with the uninfected group as a control, indicated that MDMs infected with HIV-2 B5, B7, or Rod or with HIV-1 Ada produced significantly higher amounts of M-CSF than uninfected cells (Figure 3). In contrast, when cells infected with HIV-1 Ada were used as the control group for multiple comparisons, no significant difference was observed between HIV-1 Ada and each of the HIV-2 virus isolates. Of note, additional HIV isolates, including HIV-2 B4 and the minimally passaged HIV-1 isolates 92UG024 and BCF03, were also used for infection of MDMs from some, but not all, of the donors due to the limited size of the virus stocks. Like HIV-1 Ada, HIV-1 92UG024 and HIV-1 BCF03 also infected and replicated robustly in MDMs, which led to induction of M-CSF production (Appendix A). Statistical analysis of the data from infections with the additional individual HIV isolates by one-way ANOVA with Dunnett’s test for multiple comparisons indicated that M-CSF production by infected MDMs did not differ significantly across HIV isolates (Appendix A). Taken together, these results indicate that infection of human MDMs with both HIV-1 and HIV-2 can induce M-CSF production.

The kinetics and the amounts of M-CSF production varied among the HIV isolates and donors tested. The amount of M-CSF production (Figure 2B,D) did not always correlate with the level of virus replication (Figure 2A,C). The amounts of M-CSF were often substantial (over 2-fold) even when HIV-2 replication was relatively low (HIV-2 B8 and Rod, Figure 2C). This observation is consistent with our previous studies using HIV-1 whereby maximal M-CSF production could be observed even at lower thresholds of virus replication [46]. However, for each HIV isolate, the replication kinetics correlated with M-CSF production by MDMs during the infection process (Figure 4). The correlation coefficients of HIV-1 Ada and HIV-2 B5, B7, B8, and Rod isolates were 0.788 ± 0.003, 0.722 ± 0.007, 0.770 ± 0.004, 0.629 ± 0.011, and 0.671 ± 0.013, respectively (Figure 4F). These data indicate a moderate (R = 0.5–0.7) to high (R = 0.7–0.9) correlation between the degree of HIV virus replication and M-CSF production.

### 2.3. MDMs Infected with HIV-1 and HIV-2 Isolates Produce Different Levels of CXCL7

The similar abilities of HIV-1 and HIV-2 isolates to induce M-CSF production following their infection of MDMs suggest that M-CSF is unlikely to be a factor associated with their different pathogenicity. Recently, we revealed that infection of MDMs with HIV-1 or HIV-2 differentially regulates production of the β-chemokine, CCL2. To further investigate whether other chemokines can also be differentially regulated during infection with HIV-1 or HIV-2, we compared the expression profiles of 12 chemokines in MDMs following infection with HIV-1 Ada or HIV-2 B5, B9, or Rod isolates by Affymetrix microarray, using cell samples prepared at the peak of viral replication. Among the 12 chemokines, the pro-platelet basic protein (PPBP) gene was identified as upregulated in HIV-1-infected MDMs but downregulated in HIV-2-infected MDMs (Figure 5A). The protein encoded by *PPBP* is a platelet-derived growth factor that belongs to the CXC chemokine family and can be cleaved to produce the chemokine, CXCL7 [59,60]. The differences in regulation of PPBP/CXCL7 gene expression were confirmed in MDMs infected with HIV-1 Ada or Bal or HIV-2 B5, B7, B8, or Rod isolates by qPCR (Figure 5B).

To further corroborate this finding, we next measured the concentrations of CXCL7 protein in HIV-infected MDM culture supernatants that were collected from day 6 to day 36 at 6-day intervals by ELISA. Similarly to M-CSF production, the kinetics and the amounts of CXCL7 production varied among the HIV isolates and donors tested, and the amounts of CXCL7 production did not always correlate with the amount of virus replication (Figure 6A,B). However, when compared with uninfected MDMs, MDMs infected with HIV-1 secreted higher amounts of CXCL7, while MDMs infected with HIV-2 isolates produced lower amounts of CXCL7 during the course of infection (Figure 6B). Statistical analysis of CXCL7 production at the time of peak virus replication in MDMs from the six to eight donors by one-way ANOVA using Dunnett’s test for multiple comparisons indicated that the relative amounts of CXCL7 produced by MDMs infected with individual HIV isolates were significantly different (*p* < 0.0001) (Figure 6C). Multiple comparisons by Dunnett’s test using the HIV-1 Ada group as a control indicated that MDMs infected with individual HIV-2 B5, B7, or B8 isolates produced significantly lower amounts of CXCL7 (Figure 6C). In addition, statistical analysis of the CXCL7 levels in the supernatants harvested from infection of MDMs with additional HIV isolates (described in Section 2.2) further indicates that MDMs produced different levels of CXCL7 following infection with HIV-1 or HIV-2 (Appendix A). Together, these results demonstrated that HIV-1 and HIV-2 infection of human MDMs differentially regulate production of the chemokine CXCL7.

## 3. Discussion

In the current study, we established that in vitro infection of primary human MDMs with HIV-1 or HIV-2 induced the production of similar levels of the cytokine M-CSF but had different effects on CXCL7 chemokine expression. The increased M-CSF production was consistent with and further supports previous observations regarding the role of M-CSF in HIV-1 infection [46,48,49,55,61]. However, induction of M-CSF production does not appear to be a universal feature of MDMs responding to virus infections. Other viruses capable of replicating in human MDMs, including measles and respiratory syncytial viruses, as shown in one of our previous studies, did not enhance the production of M-CSF but did have the capacity to induce the production of pro-inflammatory cytokines or chemokines [55]. In contrast, M-CSF and IL-34, which facilitate macrophage survival and differentiation and mediate their biological activity via binding to and signaling through the M-CSF receptor (c-fms/CD115), are produced at high levels in chronic hepatitis C virus infection [62]. Although Epstein-Barr virus (EBV) has not been reported to induce production of M-CSF, it expresses an early gene product (BARF1) that contributes to viral pathogenesis by forming a complex with M-CSF, causing a reduction in M-CSF biological activity [63]. In addition, Kaposi’s sarcoma herpes virus (KSHV) has been shown to encode proteins capable of interfering with the development of macrophages mediated by M-CSF that are essential for their differentiation and survival [64]. These findings indicate the complex role that M-CSF may play in the pathogenesis of different viruses via increases in its production or interference with its biological activity.

Although the ability of HIV-2 to infect human macrophages has previously been reported [40], detailed investigations regarding the effect of this virus on M-CSF production by MDMs have not been conducted. Using the minimally passaged HIV-2 isolates B5, B7, B8, and Rod, we found that all tested HIV-2 isolates had the capacity to replicate in human MDMs (Figure 1, Figure 2 and Figure 6) and induce the production of M-CSF (Figure 2 and Figure 3). The levels of M-CSF were increased compared to those in uninfected MDMs and comparable to those induced by HIV-1 infection but did not always correlate with the level of virus replication (Figure 2). We also observed that even low levels of HIV replication were capable of inducing M-CSF expression. Furthermore, we found that for minimally passaged HIV-2 isolates, the co-receptor used appeared to have little impact on the ability of the virus to induce M-CSF production (Figure 2 and Figure 3). The complex mechanism(s) of HIV-induced M-CSF production in MDMs is still largely undefined. Our early studies revealed that virion binding to MDMs alone did not induce production of M-CSF and that HIV-1 binding, entry, and replication were all required [46]. In addition to HIV-1 replication, cell-to-cell interactions between macrophages and lymphocytes may further augment M-CSF production [65,66]. Our preliminary attempts to identify the HIV-1 gene(s) responsible for induction of M-CSF in human MDMs using molecularly cloned deletion mutants showed that removal of HIV-1 Vpr, Vpr/Vpu, or Vif reduced virus replication, but the impact on M-CSF production could not be definitively ascertained when the results from several human donors were compared, most likely because impaired replication of the mutated viruses failed to induce adequate levels of M-CSF for comparison. Although it was shown that HIV-1 Nef may affect the production of several pro-inflammatory cytokines and chemokines, no meaningful impact on the induction of M-CSF was reported [67,68,69]. However, there is evidence that Nef may interfere with M-CSF receptor signaling via activation of Hck tyrosine kinase [70].

M-CSF promotes HIV infection of macrophages via multiple mechanisms [61]. These include increased expression of CD4, CCR5, and CXCR4 receptors to facilitate virus entry [49,71,72,73]; reduced susceptibility of HIV-1 entry to inhibition by β-chemokines such as CCL5 (RANTES) [74]; enhancement of viral gene expression [71]; offsetting of the effects of cytokines like IL-32 that suppress HIV-1 replication [65]; and promotion of the ontogeny and survival of macrophages, contributing to both the number and longevity of the infected cells. Our findings that M-CSF production was a common feature of HIV-1 and HIV-2 infection imply a greater role of M-CSF in the establishment and maintenance of lentivirus infection, including a role in the re-activation of virus reservoirs [75], and the above effects may be associated with the ability of M-CSF to facilitate macrophage differentiation into an M2 phenotype [65,66,75]. However, the M2 phenotype encompasses multiple subtypes, as macrophage differentiation may be affected by a range of cytokines and maturation factors. Furthermore, the macrophage population in vivo likely includes a spectrum of cells with features that place them between the M1 or M2 phenotypes induced in vitro [52]. Being aware of the potential deficiencies of any in vitro model in mimicking real life scenarios, we used pooled human serum from multiple donors in an attempt to more adequately replicate the cytokine milieu naturally encountered by monocytes and generate cells resembling the macrophage population encountered by HIV-1 and/or HIV-2 in vivo [76].

It is still not well understood why HIV-2 is less pathogenic than HIV-1. The similar behavior of HIV-1 and HIV-2 in the induction of M-CSF excludes this cytokine as a potential factor contributing to the observed differences in their pathogenicity. Acute HIV infection also induces production of chemokines (CCL2, CCL3, CCL4, and CCL5), which may recruit uninfected monocytes/macrophages and/or CD4^+^ T lymphocytes to the sites of virus replication, thus promoting the spread of infection and/or the establishment of virus reservoirs [20,77,78]. Recently, we revealed that CCL2 expression was increased in HIV-1-infected MDMs but decreased in HIV-2-infected MDMs compared to uninfected controls [41]. The significance of CCL2 as a factor associated with the severity of HIV infection was further supported by the observation that CCL2 expression was enhanced in viremic (>100,000 RNA copies/mL) compared to aviremic (<50 RNA copies/mL) HIV-1-positive individuals [79].

In this study, we identified CXCL7 as an additional chemokine that is differentially regulated in human MDMs infected with HIV-1 versus HIV-2. CXCL7, also known as neutrophil activating peptide 2 (NAP2), is an ELR^+^ chemokine originally identified in activated platelets and later found to be expressed in a variety of cells, including monocytes/macrophages [60,80,81,82]. CXCL7 is synthesized as a pro-platelet basic protein (PPBP) precursor and undergoes proteolytic cleavage to generate the active chemokine [60,83]. The potential mechanisms responsible for the differential regulation of CXCL7 during HIV-1 and HIV-2 infection remain to be elucidated. The observed changes in the mRNA levels of the CXCL7 precursor PPBP (Figure 5) suggest that differences in NF-κB regulation during HIV-1 and HIV-2 infection might play a role [84].

CXCL7 is potent chemotactic agent and appears to be an important factor in tumorigenesis and inflammation that acts by recruiting cells of the immune system, although the precise mechanisms underlying CXCL7’s pleotropic cellular effects remain to be elucidated [60,85,86]. At present, there is very limited information regarding the potential role of CXCL7 in the pathogenesis of HIV infection. Our pilot study suggests that CXCL7 may play a pivotal role by attracting lymphocytes (Appendix A) to the sites of virus replication, thus enhancing the spread of infection. Our hypothesis is consistent with the observations of Landrø et al., who found that CXCL7 plasma levels in vivo were significantly higher in HIV-infected (presumably HIV-1-infected) patients before and during highly active antiretroviral therapy (HAART) compared to healthy controls [87]. Another study also reported increased CXCL7 levels in HIV-1 infected individuals regardless of their clinical status, but the difference from uninfected controls was not statistically significant [88]. Significantly increased CXCL7 levels were also found in a small group of long-term non-progressors (LTNP) compared to healthy controls [89]. Unfortunately, this study did not include patients with typical or accelerated progression to AIDS. The study also revealed that the LTNP had high titers of HIV-1 neutralizing antibodies, which likely offset the infection-promoting effect of CXCL7 [89].

Based on the release of cytokines and chemokines following acute HIV infection, we have proposed a model wherein chemokines produced by HIV-infected macrophages attract uninfected monocytes/macrophages and/or lymphocytes to the sites of virus replication. M-CSF secreted by the infected cells then renders the incoming monocytes/macrophages more susceptible to infection by facilitating cell differentiation, enhancing expression of CD4, CCR5, and/or CXCR4 and potentially affecting virus replication after entry. Reduced expression of CCL2 and CXCL7 chemokines following HIV-2 infection may impair recruitment of macrophages and/or T cells to the sites of virus replication and hence impede virus spread and/or establishment of viral reservoirs. In addition, decreased CCL2 and CXCL7 expression may also contribute to the less pronounced activation of the immune system during HIV-2 infection compared to HIV-1 infection [16]. Whether or not CXCL7 has an additional role in regulating HIV pathogenesis, as by impacting activation of the coagulation system, remains to be determined.

In conclusion, our current and previous [41] studies reveal that infection of MDMs with HIV-2 induced the production of similar levels of M-CSF but lower levels of CXCL7 and CCL2 compared to MDMs infected with HIV-1. These findings imply that M-CSF may play a similar role in supporting HIV-1 and HIV-2 infection and subsequent virus replication, while differential regulation of certain chemokines following infection with HIV-1 and HIV-2 may correlate with their distinct pathogenicity. Finally, revealing the cytokine patterns during HIV-1 and/or HIV-2 infection of primary human macrophages may provide important information to support the development of better in vitro assays and thereby facilitate the characterization and quality testing of new potential anti-HIV therapies.

## 4. Materials and Methods

### 4.1. Monocyte Isolation and Culture

Leukapheresis blood units from healthy HIV-seronegative donors were obtained from the Department of Transfusion Medicine (DTM) under an Institutional Review Board (IRB)-approved protocol at the National Institutes of Health. A categorical exemption is in place for experimental studies performed by CDER/FDA researchers using existing deidentified blood donors who provided written informed consent according to the international ethical guidelines for biomedical research involving human subjects; these samples were anonymized prior to being sent to the FDA. Peripheral blood mononuclear cells (PBMC) were isolated from human blood following leukapheresis of HIV-1-seronegative donors and subsequent density gradient centrifugation; monocytes were purified by countercurrent centrifugal cell elutriation, as previously described [46]. Over 90% of the cells in the elutriated monocyte fraction were CD14^+^, as determined by flow cytometry analysis, and showed ≥95% viability (trypan blue exclusion test). Monocytes were differentiated in culture for 7–10 days at 37 °C in 5% CO_2_ at a concentration of 2 × 10^6^ cells/mL, 2.0 mL per well, in six-well tissue-culture plates (Costar, Cambridge, MA, USA) using DMEM (Invitrogen, Carlsbad, CA, USA) complete medium containing 10% pooled human serum (PHS), 2 mM L-glutamine, 1 mM sodium pyruvate, and penicillin (50 units/mL)/streptomycin (50 μg/mL) (Invitrogen) to generate MDMs. The PHS was generated in house from plasma from multiple non-transfused, healthy, HIV-seronegative male donors obtained from the NIH DTM.

### 4.2. HIV Infection of Human Monocyte-Derived Macrophages

After differentiation, MDMs were harvested by scraping and then replated into 24-well tissue-culture plates (Nunc, Inc., Naperville, IL, USA), at a concentration of 0.5 × 10^6^ cells/mL, 1.5 mL per well. After 24 to 48 h, MDMs were infected with individual HIV-1 Ada, Bal, 92UG024, and BCF03 isolates as previously described [46] or with primary isolates of HIV-2 that had been minimally passaged in human PBMC. HIV-1 Bal was purchased from Advanced Biotechnologies (Columbia, MD, USA). HIV-1 Ada was obtained through the AIDS Research and Reference Reagent Program (ARRRP), NIAID, NIH, from Dr. H. Gendelman [47,90], then expanded in human macrophages prior to purification by ultracentrifugation and cryopreservation (Advanced Biotechnologies) [46]. HIV-1 92UG024 (Cata#1649) and HIV-1 BCF03 (Cata#3335) are minimally passaged HIV-1 isolates that were obtained from the AIDS Research and Reference Reagent Repository. HIV-2 B2-B5 and HIV-2 B7-B9 were described previously [56], and the stocks of these HIV-2 isolates were expanded as previously described [46]. An early PBMC-passaged, non-cloned isolate of HIV-2 Rod that was obtained from Dr. F. Clavel in 1987 [91] and cryopreserved was cultured in human MDMs to generate the virus stock used in these studies. Input virus for the primary HIV-2 isolates was equalized based on reverse transcriptase (RT) activity using 0.1 cpm/cell or HIV-2 p26 (5 ng/well). HIV-1 input was also normalized based on RT activity. Virus adsorption was carried out for 4 h at 37 °C. Cells were then washed, and fresh medium was added. Every 3 days post-infection, 80% of the culture medium was collected, stored at −80 °C, and replaced with fresh medium. Subsequently, supernatants were assessed for cytokine/chemokine at 6-day intervals.

### 4.3. GHOST Cell Assay for Determination of HIV Co-Receptor Usage

Parental GHOST cells and GHOST cell lines expressing CCR1, CCR2b, CCR3, CCR4 CXCR4, CCR8, Bonzo and CD4 were obtained through the NIH AIDS Research and Reference Reagent Program, Division of AIDS, NIAID, NIH [58]. The GHOST-CCR5 cell line was obtained from Dr. Nathaniel Landau [57,92]. These GHOST cell lines were used to determine co-receptor usage by HIV-2 isolates. Briefly, cells were seeded at a concentration of 10^5^ cells/well in a Costar 24-well tissue-culture plate in MEM medium (Quality Biological, Inc., Gaithersburg, MD, USA) prior to the day of infection. Subsequently, these cells were infected with HIV-2 B2-5, and B7-9, as well as HIV-2 Rod (5 ng p26 per well); incubated at 37 °C/5% CO_2_ for 2 h; washed thoroughly; and cultured in MEM medium supplemented with 10% FBS (Hyclone/Thermo Scientific; Logan, UT, USA); 10 μg/mL puromycin (Sigma; St. Louis, MO, USA); 100 μg/mL hygromycin (Invitrogen; Carlsbad, CA, USA); 500 μg/mL G418; penicillin/streptomycin; and L-glutamine (Quality Biological, Inc.). Culture supernatants were harvested on days 4, 8, and 12; HIV-2 virus replication was determined by measuring p26 antigen levels using the SIV p27 antigen ELISA kit (Zeptometrix Corporation; Buffalo, NY, USA) as described previously [93] after we confirmed that this ELISA kit cross-reacts with HIV-2 p26 antigen.

### 4.4. Reverse Transcriptase Assay

Progression of virus replication in MDMs infected with HIV-1 or HIV-2 was monitored by measuring RT activity. The RT assay is a ^3^H-based modification of the method described by Hoffman, et al. [46]. Values shown reflect the average of duplicate samples (cpm/25 μL) that differed by not more than 15%.

### 4.5. M-CSF Bioassay

M-CSF biological activity was determined by measuring proliferation of the M-CSF- dependent murine cell line, M-NFS-60, as previously described [46,94,95]. An Infinite F50 plate reader (Tecan, Durham, NC, USA) was used to measure absorbance. The limit of detection for the M-CSF bioassay is 1.5 ng/mL. Confirmation of M-CSF identity or verification of the concentration was performed early in these studies using neutralizing anti-M-CSF antibodies or an M-CSF-specific ELISA (R&D Systems, Minneapolis, MN, USA) [46].

### 4.6. CXCL7 ELISA

CXCL7 concentrations in the supernatants collected from cultures of HIV-infected MDMs throughout the time course of infections were measured using ELISA kits (Cat# EA100464) purchased from Origene (Gaithersburg, MD, USA), following the manufacturer’s instructions.

### 4.7. Quantitative PCR

Total RNA was extracted from cells by TRIzol reagent (Invitrogen, Carlabad, CA, USA), and 2 μg RNA was reverse transcribed to cDNA using Omniscript (Qiagen Technology, Germantown, MD, USA) and random-hexamer primer methods. CXCL7 primers and the probe were purchased from Applied Biosystems (Cat# Hs00234077, Forster City, CA, USA). Each 30 μL volume of the PCR reaction included 2 μL 2× diluted cDNA (equal 100 ng RNA), 300 nM sense and antisense primers, 200 nM 6′-Fam labeled fluorescence target probe, and 1.5 μL 20× glyceraldehyde-3-phosphate dehydrogenase (*GAPDH*) endogenous control. Amplification of an endogenous control was performed in the same wells to standardize the level of expression of the target RNA. The CT data from real-time PCR was used to determine ΔCT and ΔΔCT for the relative amount of the target mRNA. The formula 2^−ΔΔCT^ was used to calculate the level of CXCL7 gene expression. The ΔCT as obtained by subtracting the *GAPDH* CT value from the CXCL7 gene CT value. The ΔΔCT was obtained by subtracting the control ΔCT from target ΔCT.

### 4.8. Statistical Analysis

Statistical analysis of data from more than two groups was performed using the one-way ANOVA with Dunnett’s multiple comparisons test that is built into Prisms GraphPad Version 10 (Dotmatics, Boston, MA, USA). Statistical analysis of data between two groups was performed using Student’s *t*-test. The statistical significance of any difference was determined by a *p*-value < 0.05.

## Figures and Tables

**Figure 1 ijms-26-05028-f001:**
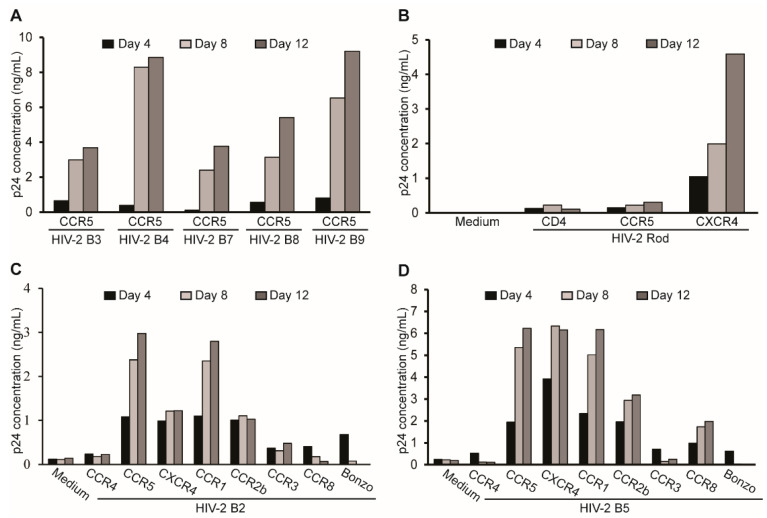
Co-receptors used by HIV-2 isolates for infection. GHOST cell lines expressing the coreceptors CCR5 (**A**); CD4, CCR5, or CXCR4 (**B**); CCR4, CCR5, CXCR4, CCR1, CCR2b, CCR3, CCR8, or Bonzo (**C**,**D**) were infected with the HIV-2 isolates indicated in the figure. HIV-2 p26 antigen levels in supernatants harvested from HIV-2-infected GHOST cells were measured at the time points indicated using an SIV p27 ELISA kit that also detects HIV-2 p26 antigen.

**Figure 2 ijms-26-05028-f002:**
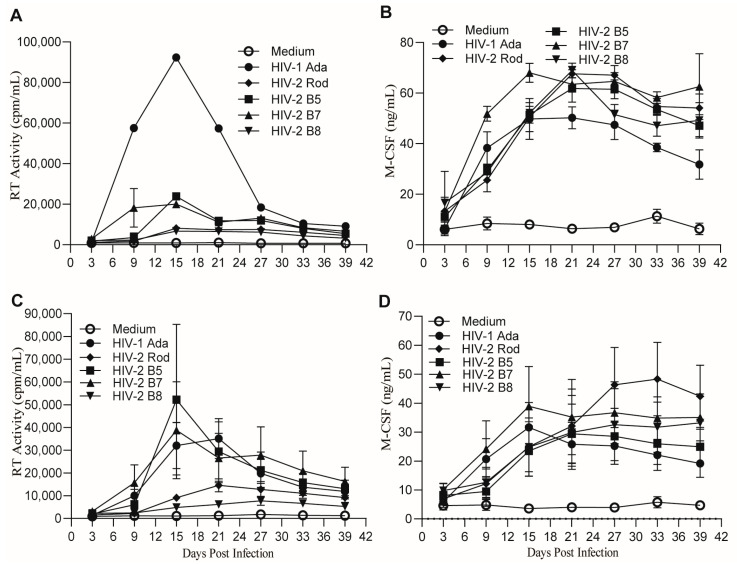
Time course of virus replication and M-CSF production following infection of MDMs with HIV-1 or HIV-2 isolates. MDMs differentiated for 10 days in complete DMEM containing 10% PHS were harvested, replated, and infected with the HIV-1 isolate Ada, the pleiotropic HIV-2 isolate B5, the CXCR4-tropic HIV-2 Rod, or the CCR5-tropic HIV-2 isolates B7 or B8. Culture supernatants were collected from day 3 to day 39 at 3-day intervals, and cultures were provided with fresh medium following supernatant collection. Harvested supernatants were assessed every 6 days for virus replication by measuring RT activity (**A**,**C**), and M-CSF biological activity (**B**,**D**) by measuring proliferation of the M-CSF-dependent murine cell line M-NFS-60. Data shown in (**A**,**B**) are results for HIV virus replication and M-CSF production from representative donor #5. Data shown in (**B**,**D**) are results from five donors (Mean ± SEM, *n* = 5).

**Figure 3 ijms-26-05028-f003:**
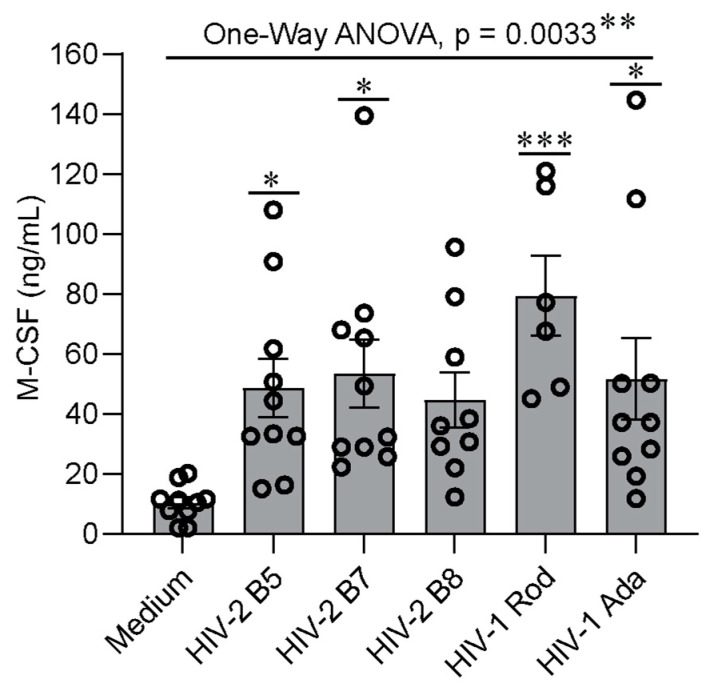
M-CSF production by MDMs is increased following infection with HIV-1 or HIV-2 strains. Supernatants were harvested from MDMs that were uninfected (Medium) or MDMs that were infected in parallel with individual HIV-2 isolates (B5, B7, B8, Rod, or HIV-1 Ada) at the time of peak viral replication. The levels of M-CSF present in the supernatants were measured by M-CSF-dependent proliferation of M-NFS-60 cells. Each symbol represents one individual donor. The asterisks on the long bar line indicate significant differences between the uninfected and the five HIV-1 and HIV-2 infection groups (**, *p* < 0.005, *n* = 6–10), as was analyzed using one-way ANOVA. The asterisks on the short bar lines indicate significant differences (*, *p* < 0.05 and ***, *p* < 0.0005) between the HIV-infected group under the bar lines and the uninfected control group, as assessed using Dunnett’s test for multiple comparisons.

**Figure 4 ijms-26-05028-f004:**
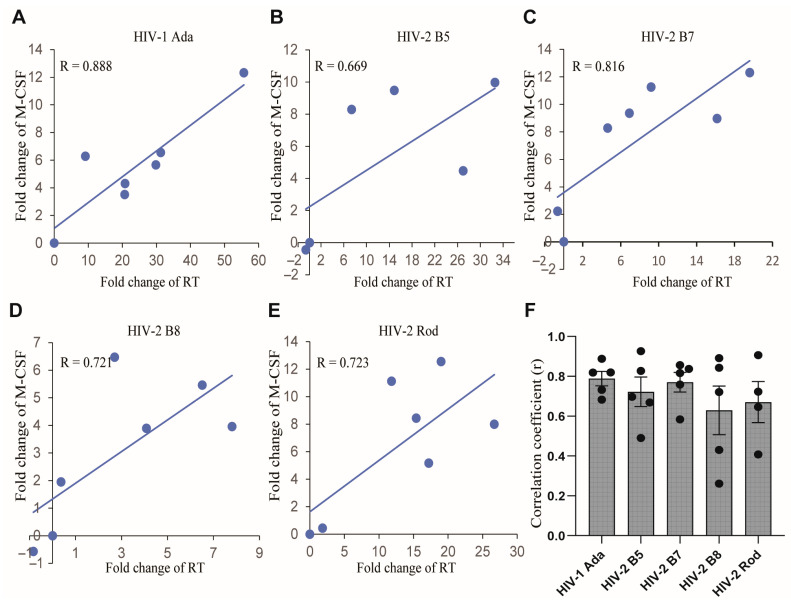
Correlation between HIV virus replication and M-CSF production. RT activities and M-CSF levels in the culture supernatants collected from day 3 to day 39 at 6-day intervals were used for generating the scatterplots and correlation coefficients by Excel. Panels (**A**–**E**) show the scatterplots and the correlation coefficients (R) of HIV-1 Ada and HIV-2 isolates B5, B7, B8, and Rod from one representative donor. Data shown in Panel (**F**) are correlation coefficients of the HIV isolates from 4–5 donors, with each symbol representing one individual donor.

**Figure 5 ijms-26-05028-f005:**
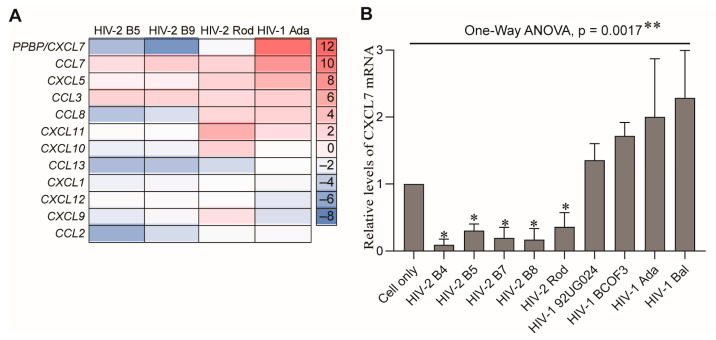
Differential regulation of CXCL7 (PPBP) gene expression in MDMs infected with HIV-1 or HIV-2. (**A**) The expression profiles of chemokine genes in MDMs infected with HIV-1 Ada or HIV-2 B5, B9, or Rod was analyzed by Affymetrix array using cell samples collected on day 15 post-infection. (**B**) The expression levels of CXCL7 mRNA in MDMs infected with HIV-1 Ada, 92UG024, BCF03, or Bal or HIV-2 B4, B5, B7, B8, or Rod isolates were measured by qPCR. The expression levels of CXCL7 mRNA were examined using cells harvested on day 15 post-infection, normalized by the levels of GAPDH mRNA and presented as the amount relative to uninfected cells (Cell only). Statistical analysis was performed using one-way ANOVA with Dunnett’s test for multiple testing corrections. The asterisks on the long bar line indicate a significant difference between the uninfected group and the nine HIV-1 and HIV-2-infected groups (**, *p* < 0.005, *n* = 3), as found using one-way ANOVA. The asterisks on the short bar lines indicate significant differences (*, *p* < 0.05) between the HIV infected group under the bar lines and the HIV-1 Ada control group, as assessed using Dunnett’s test for multiple comparisons.

**Figure 6 ijms-26-05028-f006:**
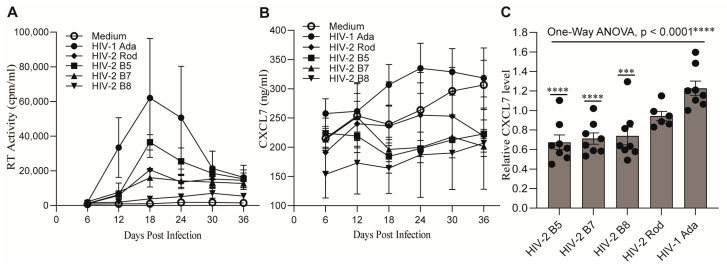
Differential CXCL7 production by MDMs following infection with HIV-1 or HIV-2. (**A**,**B**) Time course of virus replication and CXCL7 production following infection of MDMs with HIV-1 or HIV-2 isolates. MDMs differentiated for 10 days in DMEM containing 10% PHS were harvested, replated, and infected with the indicated HIV isolates. RT activities (**A**) and CXCL7 levels (**B**) of the culture supernatants collected from day 6 to day 36 were assessed at 6-day intervals. Data shown are results from three donors (Mean ± SEM, *n* = 3). (**C**) The concentrations of CXCL7 in the supernatants harvested from MDMs infected with the indicated HIV isolates at the peak viral-replication time points were determined by ELISA. The concentrations of CXCL7 in the supernatants of each infection group were normalized to the expression levels of CXCL7 in the supernatants harvested from uninfected MDMs and presented as relative CXCL7 levels. Each symbol represents an individual donor. Data are shown as the mean ± SEM (*n* = 6–8). The asterisks on the long bar line indicate a significant difference among the five HIV-infection groups (*p* < 0.0001, *n* = 6–8), as analyzed using one-way ANOVA. The asterisks on the short bar lines indicate significant differences (***, *p* < 0.0005, ****, *p* < 0.0001) between the HIV-infection groups under the bar lines and the HIV-1 Ada control group, as assessed using Dunnett’s test for multiple comparisons.

**Table 1 ijms-26-05028-t001:** HIV-2 isolates from Portugal used in this study.

Isolate	Origin	Co-Receptor ^1^
HIV-2 B2	Portugal	CXCR4, CCR5, CCR1, CCR2B
HIV-2 B3	Portugal	CCR5
HIV-2 B4	Portugal	CCR5
HIV-2 B5	Portugal	CXCR4, CCR5, CCR1, CCR2B
HIV-2 B7	Portugal	CCR5
HIV-2 B8	Portugal	CCR5
HIV-2 B9	Portugal	CCR5

^1^ Co-receptor: Chemokine receptor that acts in concert with CD4 to enable HIV–cell fusion.

## Data Availability

The data presented in this study are available in this article in the main figures or tables.

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
