# Peer review of "Macrophage-Derived Factors with the Potential to Contribute to the Pathogenicity of HIV-1 and HIV-2: Roles of M-CSF and CXCL7"

_ijms, 2025, doi:10.3390/ijms26115028_

Round 1
Reviewer 1 Report
Comments and Suggestions for Authors
The manuscript by Gao et al. focuses on comparing the pathogenicity of HIV-1 and HIV-2 infection in the context of cytokine expression, specifically M-CSF and CXCL7. The authors compared infection of several primary HIV-1 isolates with the HIV-1 laboratory isolates ADA and Bal in primary human monocyte-derived macrophages (MDMs) and found that while the HIV isolates replicated in MDMs with different capacities, they induced similar amounts of host cell-produced M-CSF. However, among the various chemokines tested, CXCL7 displayed significantly higher levels in HIV-1-infected MDMs, whereas HIV-2 replication correlated with reduced CXCL7 production even compared to uninfected cells, suggesting that HIV-2 infection downregulates CXCL7, which may be a factor in the lower pathogenicity of HIV-2.
The manuscript is well-written and logically composed. However, despite the fairly clear experimental results that allow drawing preliminary conclusions about the pathogenicity of various types of HIV, a comparison of laboratory, cell culture-optimized, and high-fit macrophage-tropic HIV-1 strains (both from the highly pathogenic B subtype of the virus) with primary low-passage isolates of HIV-2 does not appear methodologically correct. To validate the results, infection of MDM with primary HIV-1 isolates from subtype B and other subtype(s) prevalent in Africa (HIV-1 CRF02_AG, prevalent in West Africa may be the most interesting) should be compared with HIV-2 in the same experiments.
Below I point out a few minor items that should also be addressed by the authors to improve the manuscript in some details.
- In abstract, the authors state that to investigate the potential role of cytokines, they compared the expression levels of M-CSF and CXCL7. It ‘s not clear why they selected these particular cytokines for analysis. It is important to preface this statement with the information that chemokine profiling revealed that CXCL7 was significantly correlated with the type of HIV with which the cells were infected.
- In the Materials and Methods section, the authors describe how they differentiated elutriated monocytes from PBMCs toward MDMs. No G-CSF or GM-CSF were added. Without these growth factors, monocytes form a population of macrophage-like cells with a heterogeneous phenotype that in many cases depends on the specific donor. How can the authors ensure that the same cells were infected with the viruses and that the results are comparable?
- In subsection 2.3 the authors indicate that HIV-2 for infection was p24 normalized (5 ng/well), and below that viral replication was determined by measuring p24 antigen using the SIV p27 ELISA kit. Do the authors mean they normalized HIV-2 to its p26 capsid protein? Which ELISA was used to normalize HIV-1?
Author Response
We appreciate the thoughtful review of our manuscript entitled, “Macrophage-derived factors with the potential to contribute to pathogenicity of HIV-1 and HIV-2: Role of M-CSF and CXCL7” that enhanced the overall quality of the manuscript. In response to the referee’s comments, we have revised the manuscript. The following is a point-by-point response to the referee’s comments:
Comments and Suggestion for Authors:
The manuscript by Gao et al. focuses on comparing the pathogenicity of HIV-1 and HIV-2 infection in the context of cytokine expression, specifically M-CSF and CXCL7. The authors compared infection of several primary HIV-1 isolates with the HIV-1 laboratory isolates ADA and Bal in primary human monocyte-derived macrophages (MDMs) and found that while the HIV isolates replicated in MDMs with different capacities, they induced similar amounts of host cell-produced M-CSF. However, among the various chemokines tested, CXCL7 displayed significantly higher levels in HIV-1-infected MDMs, whereas HIV-2 replication correlated with reduced CXCL7 production even compared to uninfected cells, suggesting that HIV-2 infection downregulates CXCL7, which may be a factor in the lower pathogenicity of HIV-2.
The manuscript is well-written and logically composed. However, despite the fairly clear experimental results that allow drawing preliminary conclusions about the pathogenicity of various types of HIV, a comparison of laboratory, cell culture-optimized, and high-fit macrophage-tropic HIV-1 strains (both from the highly pathogenic B subtype of the virus) with primary low-passage isolates of HIV-2 does not appear methodologically correct. To validate the results, infection of MDM with primary HIV-1 isolates from subtype B and other subtype(s) prevalent in Africa (HIV-1 CRF02_AG, prevalent in West Africa may be the most interesting) should be compared with HIV-2 in the same experiments.
Response: We appreciate the thoughtful comment of the reviewer. Despite our limited access to minimally passaged HIV-1 isolates, we did perform experiments with the primary HIV-1 isolates BCF03 (Group O) and 92UG024 (Group M, Clade D, one of the few primary isolates using CXCR4 as a coreceptor and capable of replicating in macrophages), in addition to the lab adapted HIV-1 Ada and HIV-1 Bal (both Group M, Clade B). Due to limited amounts of the minimally passaged strains, we could not test the same number of donors as with the lab adapted strains and for this reason, we initially did not provide the results in our manuscript. Recognizing the importance of the experiments with minimally passaged isolates, we included these data in the revised manuscript (supplementary Figure S1, supplementary Figure S2 and supplementary Table 1). The supplementary data represent results from MDM derived from three donors infected with HIV-1 92UG024 or HIV-1 BCF03, in parallel with individual HIV-1 Ada and HIV-2 isolates. The results were statistically analyzed by the One-Way ANOVA with the Dunnett’s test for multiple testing corrections and are consistent with the M-CSF and CXCL7 findings established using HIV-1 Ada and HIV-1 Bal. We were unable to obtain HIV-1 CRF02_AG, but performed experiments with a strain of HIV-1 Group O (HIV-1 BCF03), which is also predominantly found in West and Central Africa. In summary, we think that the group of HIV-1 and HIV-2 isolates used in the current study is sufficiently diverse to support our conclusion regarding the role of M-CSF and CXCL7 expression during HIV-1 and HIV-2 infection.
Below I point out a few minor items that should also be addressed by the authors to improve the manuscript in some details.
- In abstract, the authors state that to investigate the potential role of cytokines, they compared the expression levels of M-CSF and CXCL7. It‘s not clear why they selected these particular cytokines for analysis. It is important to preface this statement with the information that chemokine profiling revealed that CXCL7 was significantly correlated with the type of HIV with which the cells were infected.
Response: We thank the reviewer for this comment. Our previous studies show that M-CSF is a cytokine playing a role in promoting HIV infection and replication. In our pilot gene expression study that compared the gene expression profiles of MDM infected with HIV-1 or HIV-2, the chemokine CXCL7 was initially identified to be expressed in correlation with HIV type. This information was added to the revised abstract. Because of word limitation, we also revised the abstract accordingly to meet the requirement of no more than 200 words in the abstract.
- In the Materials and Methods section, the authors describe how they differentiated elutriated monocytes from PBMCs toward MDMs. No G-CSF or GM-CSF were added. Without these growth factors, monocytes form a population of macrophage-like cells with a heterogeneous phenotype that in many cases depends on the specific donor. How can the authors ensure that the same cells were infected with the viruses and that the results are comparable?
Response: We presume the reviewer had in mind M-CSF instead of G-CSF. We are aware that M-CSF, GM-CSF or a combination of both are widely used in monocyte differentiation in vitro to improve yield. However, the concentration(s) of M-CSF and/or GM-CSF typically used (≥20 ng/mL) are much higher relative to the levels of these cytokines observed in vivo. During monocyte differentiation, we used pooled human serum from multiple donors which, in our opinion, more adequately resembles the cytokine milieu encountered by monocytes in vivo. Furthermore, it is more likely that HIV-1 or HIV-2 will encounter a more heterogeneous macrophage population in vivo [1], compared to the M1 or M2 macrophage phenotypes induced in vitro by a high concentration of GM-CSF or M-CSF, respectively. Also, there is a possibility that pre-incubation of monocytes/macrophages with nonphysiologically high M-CSF concentrations may affect the HIV-1 and/or HIV-2 induced M-CSF secretion. Finally, our approach is consistent with previous studies demonstrating monocyte to macrophage differentiation without the use of exogenous growth factors [2, 3].
- In subsection 2.3 the authors indicate that HIV-2 for infection was p24 normalized (5 ng/well), and below that viral replication was determined by measuring p24 antigen using the SIV p27 ELISA kit. Do the authors mean they normalized HIV-2 to its p26 capsid protein? Which ELISA was used to normalize HIV-1?
Response: We thank the reviewer for having caught this typo. The use of SIV p27 ELISA kit to measure the HIV-2 p26 protein was previously described [4]. By sequence alignment, HIV-2 p26 and SIV p27 exhibit 91% homology, supporting the cross-reaction of HIV-2 p26 with the SIV p27 ELISA kit. In addition, the cross-reaction of HIV-2 p26 with the SIV p27 ELISA kit was confirmed in our lab prior to using the SIV p27 ELISA kit for measuring HIV-2 replication. The typos were corrected in the revised manuscript. For HIV-1, virus input was normalized based on the reverse transcriptase (RT) activity, which was clarified in the Materials and Methods of the revised manuscript.
- Sattentau, Q. J.; Stevenson, M., Macrophages and HIV-1: An Unhealthy Constellation. Cell Host Microbe 2016, 19, (3), 304-10.
- Bergamini, A.; Perno, C. F.; Dini, L.; Capozzi, M.; Pesce, C. D.; Ventura, L.; Cappannoli, L.; Falasca, L.; Milanese, G.; Caliò, R.; et al., Macrophage colony-stimulating factor enhances the susceptibility of macrophages to infection by human immunodeficiency virus and reduces the activity of compounds that inhibit virus binding. Blood 1994, 84, (10), 3405-12.
- Fantuzzi, L.; Canini, I.; Belardelli, F.; Gessani, S., HIV-1 gp120 stimulates the production of beta-chemokines in human peripheral blood monocytes through a CD4-independent mechanism. J Immunol 2001, 166, (9), 5381-7.
- Devadas, K.; Biswas, S.; Haleyurgirisetty, M.; Ragupathy, V.; Wang, X.; Lee, S.; Hewlett, I., Identification of Host Micro RNAs That Differentiate HIV-1 and HIV-2 Infection Using Genome Expression Profiling Techniques. Viruses 2016, 8, (5).

Reviewer 2 Report
Comments and Suggestions for Authors
1# The study relies on in vitro experiments using primary human monocyte-derived macrophages (MDM). While this model is relevant, it does not fully replicate the complexity of in vivo HIV infection. The manuscript may discuss the potential limitations of using in vitro models and how these might affect the generalizability of the findings.
2# The manuscript does not provide a detailed discussion of the statistical methods used or the assumptions underlying these methods. Additionally, the manuscript does not address potential issues such as multiple testing corrections, which could affect the reliability of the p-values reported.
3# The authors may want to provide a detailed discussion of the biological mechanisms underlying the differential regulation of CXCL7. While the findings are novel, the manuscript does not explore how these differences might translate into clinical outcomes or contribute to the observed differences in pathogenicity between HIV-1 and HIV-2.
4# The study compares the effects of HIV-1 and HIV-2 on M-CSF and CXCL7 expression. However, it does not provide a detailed comparison with other related studies. A more thorough comparative analysis would help contextualize the study’s findings within the existing literature.
5# The study concludes that M-CSF may play a similar role in supporting HIV-1 and HIV-2 infection, while differential regulation of CXCL7 may correlate with their distinct pathogenicity. But according to the results, there was no enough evidence to support this claims. The relevance is not equal to causal effect. The author should better to revise their conclusion.
While the study provides valuable insights into the role of M-CSF and CXCL7 in HIV-1 and HIV-2 infections, it lacks a clear articulation of its novel contributions and detailed exploration of the biological mechanisms and clinical implications. These limitations collectively suggest that the manuscript, in its current form, would benefit from a round of revision before going public.
Author Response
We appreciate the thoughtful review of our manuscript entitled, “Macrophage-derived factors with the potential to contribute to pathogenicity of HIV-1 and HIV-2: Role of M-CSF and CXCL7” that enhanced the overall quality of the manuscript. In response to the referee’s comments, we have revised the manuscript. The following is a point-by-point response to the referee’s comments:
Comments and Suggestion for Authors:
- The study relies on in vitro experiments using primary human monocyte-derived macrophages (MDM). While this model is relevant, it does not fully replicate the complexity of in vivo HIV infection. The manuscript may discuss the potential limitations of using in vitro models and how these might affect the generalizability of the findings.
Response: We thank the reviewer for this comment and agree that an in vitro model always has its limitation. We added a relevant paragraph to the Discussion section of the revised manuscript to recognize this issue.
- The manuscript does not provide a detailed discussion of the statistical methods used or the assumptions underlying these methods. Additionally, the manuscript does not address potential issues such as multiple testing corrections, which could affect the reliability of the p-values reported.
Response: We thank the reviewer for this suggestion. We reanalyzed the data presented in Figure 3, Figure 5B and Figure 6C using the One-Way ANOVA test that is built into the Prism GraphPad. To address the potential issues of multiple testing corrections, we used the Dunnett’s method to compare every mean to a control mean (Medium group in Figure 3, HIV-1Ada infection group in Figure 5B and Figure 6C). The same One-Way ANOVA with multiple testing corrections were performed for the data presented in the supplementary figures that were generated for the revision. We revised the Materials and Methods, Figures, Figure Legends and Results accordingly.
- The authors may want to provide a detailed discussion of the biological mechanisms underlying the differential regulation of CXCL7. While the findings are novel, the manuscript does not explore how these differences might translate into clinical outcomes or contribute to the observed differences in pathogenicity between HIV-1 and HIV-2.
Response: We appreciate the reviewer’s comment and included in the Discussion of the revised manuscript an expanded paragraph reagrding the potential mechanism(s) responsible for the differences in CXCL7 expression. However, performing comprehensive experiments to study in detail these mechanisms is by itself a new project and is beyond the scope of the current study. To further support the potential clinical implications of our findings, we provide additional evidence (supplementary Figure S3) that supernatants from HIV-1 infected MDM have an increased capacity, compared to HIV-2 supernatants, to induce lymphocyte chemotaxis and that this difference is abrogated by an anti-CXCL7 antibody. In the supplementary Figure 3, the significant differences between two groups with or without anti-CXCL7 antibody were assessed by the Student’s T-Test.
- The study compares the effects of HIV-1 and HIV-2 on M-CSF and CXCL7 expression. However, it does not provide a detailed comparison with other related studies. A more thorough comparative analysis would help contextualize the study’s findings within the existing literature.
Response: Approximately 10% of all references cited in the manuscript are related to M-CSF and in general our findings are in agreement with these previous observations. However, to better address the reviewer’s comment we included additional references regarding the role of M-CSF in HIV infection, particularly in macrophage subtype differentiation, and made edits to better emphasize the consistency of our data with the previously published studies. There are very limited data regarding the role of CXCL7 in HIV-1 infection and practically no published studies related to CXCL7 and HIV-2. We included one additional reference about CXCL7 and HIV-1 [1] and discussed the potential effects of CXCL7 in the complex in vivo environment.
- The study concludes that M-CSF may play a similar role in supporting HIV-1 and HIV-2 infection, while differential regulation of CXCL7 may correlate with their distinct pathogenicity. But according to the results, there was no enough evidence to support this claims. The relevance is not equal to causal effect. The author should better to revise their conclusion.
Response: We agree with the reviewer that it is hard to establish a direct casual effect between the differential CXCL7 secretion in HIV-1 and HIV-2 infection and the differences in their pathogenicity. We would like to emphasize that we never categorically claimed such a direct association and used expressions such as “may play a role” or “may correlate”. Although it is challenging to establish a direct causal effect between CXCL7 and HIV pathogenicity, looking at the entirety of available data regarding the general role of chemokines in the function of the immune system, as well as their effects on HIV replication and the replication of other viruses in our opinion, this approach indirectly supports a potential role of CXCL7 as factor contributing to the differences in the pathogenicity of HIV-1 or HIV-2. A short paragraph was added to the Discussion regarding this issue. Also, please see our response to comment #3.
- Hao, Y.; Bai, G.; Wang, J.; Zhao, L.; Sutherland, K.; Cai, J.; Cao, C., Identifiable biomarker and treatment development using HIV-1 long term non-progressor sera. BMC Immunol 2015, 16, 25.
Round 2
Reviewer 1 Report
Comments and Suggestions for Authors
The manuscript was thoroughly revised. The authors have made requested changes to the portions of the manuscript that were criticized and added critical new data. Overall, the text, figures, and supplementary data have been substantially improved, and I found no apparent biological or methodological inaccuracies or factual gaps in the updated version. All my concerns have been addressed in the current version of the manuscript.
Reviewer 2 Report
Comments and Suggestions for Authors
All the comments have been addressed.